A pilot study: effect of irisin on trabecular bone in a streptozotocin-induced animal model of type 1 diabetic osteopathy utilizing a micro-CT

Mohsin Sahar 1 smohsin@uaeu.ac.ae
Brock Fiona 2
Kaimala Suneesh 1
Greenwood Charlene 3
Sulaiman Mohsin 1
Rogers Keith 2
Adeghate Ernest 1
1 Department of Anatomy, College of Medicine and Health Sciences, United Arab Emirates University , Al Ain, Abudhabi , United Arab Emirates
2 Cranfield Forensic Institute, Cranfield University , Shrivenham , United Kingdom
3 School of Chemical and Physical Sciences, Keele University , Newcastle-under-Lyme, Staffordshire , United Kingdom
Menini Stefano
Electronic publication date: 2023 Oct 17
Publication date: 2023
Volume: 11
Electronic Location ID: e16278
Received 2023 May 23; Accepted 2023 Sep 20
Copyright: © 2023 Mohsin et al.
Copyright year: 2023
Copyright holder: Mohsin et al.
License: This is an open access article distributed under the terms of the Creative Commons Attribution License, which permits unrestricted use, distribution, reproduction and adaptation in any medium and for any purpose provided that it is properly attributed. For attribution, the original author(s), title, publication source (PeerJ) and either DOI or URL of the article must be cited.
License URL: https://creativecommons.org/licenses/by/4.0/

Keywords: Diabetes, Irisin, Streptozotocin, Trabecular bone, Micro-CT, Osteoporosis, Diabetic osteopathy, Sclerostin, Type 1 diabetes

Funding: College of Medicine and Health Sciences, United Arab Emirates University, UAE G00003027/NP-19-14 This work was supported by the College of Medicine and Health Sciences, United Arab Emirates University, UAE, G00003027/NP-19-14, to Sahar Mohsin. The funders had no role in study design, data collection and analysis, decision to publish, or preparation of the manuscript.

==============================
Background

Osteoporosis is a significant co-morbidity of type 1 diabetes mellitus (DM1) leading to increased fracture risk. Exercise-induced hormone ‘irisin’ in low dosage has been shown to have a beneficial effect on bone metabolism by increasing osteoblast differentiation and reducing osteoclast maturation, and inhibiting apoptosis and inflammation. We investigated the role of irisin in treating diabetic osteopathy by observing its effect on trabecular bone.

Methods

DM1 was induced by intraperitoneal injection of streptozotocin 60 mg/kg body weight. Irisin in low dosage (5 µg twice a week for 6 weeks I/P) was injected into half of the control and 4-week diabetic male Wistar rats. Animals were sacrificed six months after induction of diabetes. The trabecular bone in the femoral head and neck was analyzed using a micro-CT technique. Bone turnover markers were measured using ELISA, Western blot, and RT-PCR techniques.

Results

It was found that DM1 deteriorates the trabecular bone microstructure by increasing trabecular separation (Tb-Sp) and decreasing trabecular thickness (Tb-Th), bone volume fraction (BV/TV), and bone mineral density (BMD). Irisin treatment positively affects bone quality by increasing trabecular number p < 0.05 and improves the BMD, Tb-Sp, and BV/TV by 21–28%. The deterioration in bone microarchitecture is mainly attributed to decreased bone formation observed as low osteocalcin and high sclerostin levels in diabetic bone samples p < 0.001. The irisin treatment significantly suppressed the serum and bone sclerostin levels p < 0.001, increased the serum CTX1 levels p < 0.05, and also showed non-significant improvement in osteocalcin levels.

Conclusions

This is the first pilot study to our knowledge that shows that a low dose of irisin marginally improves the trabecular bone in DM1 and is an effective peptide in reducing sclerostin levels.

Introduction

Type 1 diabetes mellitus (DM1) is associated with increased skeletal fragility, due to a decrease in both bone mineral density (BMD) and altered bone quality (Janghorbani et al., 2006; Janghorbani et al., 2007; Lebiedz-Odrobina & Kay, 2010; Napoli et al., 2017). Patients with DM1 are at greater risk of fracture due to an increasing tendency to fall not only as a result of peripheral neuropathy, poor vision, and stroke but also due to increased bone loss and/or altered bone matrix and strength (Janghorbani et al., 2006; Mohsin et al., 2019a; Vestergaard, 2007).

DM1 not only affects bone mineral density but also affects bone quality, including bone turnover, microarchitecture, mineralization, microdamage, and bone mineral composition (Hough et al., 2016; Mohsin et al., 2019a; Saito et al., 2006). Animal studies have shown changes in the bone tissue as early as 4 to 8 weeks after the onset of DM1 (Mohsin et al., 2019a). An increased number of apoptotic osteocytes were found in diabetic rat bones, which explains the imbalance of the remodeling cycle in DM1. Low levels of serum markers for bone formation such as osteocalcin and bone alkaline phosphatase and increased levels of advanced glycation end products (AGEs) were found in the streptozotocin-induced model of DM1 rats (Hygum, Starup-Linde & Langdahl, 2019; Khan & Fraser, 2015; Miyake, Kanazawa & Sugimoto, 2018). Reports on bone resorption in DM1 are particularly controversial, being reported as unchanged, decreased, or increased in animal and human population studies (Gallacher et al., 1993; Maggio et al., 2010; Motyl & McCabe, 2009; Motyl et al., 2009). The major pathogenetic mechanism involved in DM1-induced bone deficit is insulin deficiency, along with glucose toxicity, marrow adiposity, inflammation, adipokine, and other metabolic alterations (Hough et al., 2016).

Regular exercise improves the quality of life through its beneficial effects on various systems in the body. Exercise also increases bone and muscle strength and helps prevent bone loss (Benedetti et al., 2018). In turn, increasing physical activity in children with diabetes as well as good glycaemic control appears to provide some improvement in bone parameters (Colberg et al., 2016).

Irisin peptide expressed in the skeletal muscle and released after physical activity is reported to increase bone tissue mass and strength (Boström et al., 2012; Colaianni et al., 2015; Khan & Fraser, 2015). It can improve insulin resistance, lower blood glucose and promote weight loss. Studies have shown that irisin also helps in cell proliferation and inhibits cell apoptosis (Liu et al., 2017).

The role of irisin in diabetes is still unclear due to contradictory findings (Mahgoub et al., 2018). A recent study (Tentolouris et al., 2018) has shown that circulating irisin levels were lower in subjects with DM1 in comparison with healthy-matched controls. The low circulating irisin levels is associated with advanced glycation end products (AGEs) accumulation and vascular complications in diabetic patients (Rana et al., 2017), and irisin has been reported to have potent anti-inflammatory properties (Mazur-Bialy, Pocheć & Zarawski, 2017).

Browning of adipose tissue is reported with a higher irisin dose (3,500 μg kg−1 per week) but this effect was not seen with low-dose recombinant irisin (r-irisin) in young male mice (Colaianni et al., 2015). More recently it has been shown that irisin in a low dose of 100 μg kg–1 has anabolic effects on bone tissue without browning of adipose tissue. Irisin in low dose modulated the skeletal genes, Opn (osteopontin) and Sost (Sclerostin) (Colaianni et al., 2015; Holmes, 2015). Cortical bone mass and strength were markedly increased in irisin-treated mice, compared with control mice (Colaianni et al., 2015). However, this beneficial effect was only seen in cortical bone and no changes were observed in the trabecular compartment of bone in mice. A marked increase in cortical bone mass was attributed to the suppression of sclerostin which inhibits bone formation through the Wnt signaling pathway, and stimulation of ‘osteoblasts’ (bone-forming cells) (Colaianni et al., 2015). Moreover, it deters bone resorption by inhibiting osteoclast differentiation (Ma et al., 2018). Due to these actions on bone, irisin is known to enhance the mechanical properties of bone (Gallacher et al., 1993).

Trabecular bone quality is significantly lower in adults with DM1 (Shah et al., 2018) and to our knowledge, the effect of a low dose of irisin on the trabecular bone in DM has not yet been investigated. This pilot study aimed to investigate the role of a very low dose of irisin in ameliorating bone fragility associated with DM, by examining its effect on bone turnover markers and, trabecular bone microstructure using a non-destructive microcomputed tomography (micro-CT) technique in a single high-dose streptozotocin-induced model of DM1.

Materials and Methods

Animal handling, induction of diabetes, and irisin treatment

Twenty-four healthy male Wistar rats weighing between, 270 and 300 g were obtained from the Animal House Facility at United Arab Emirates University (UAEU). National Institute of Health (NIH) guidelines for the care and use of laboratory animals were followed for all experiments and procedures carried out in this study after being approved by the Animal Research Ethics Committee of the College of Medicine and Health Sciences (CMHS), UAE University ERA_2018_5833.

The animals were housed singly in cages under standard conditions with a 12 h alternating light and dark cycle, at 22–24 °C and 50–60% humidity, and provided with free access to standard rat chow and water ad libitum during the 2 weeks of acclimatization and for the experimental period. All efforts were made to minimize animal suffering and to limit the number of animals used (Mohsin et al., 2019a). No adverse event was recorded during the period of the experiment.

A single intraperitoneal (I/P) injection of streptozotocin (STZ, (U-9889) 60 mg/kg body weight; Santa Cruz Biotechnology, Dallas, TX, USA) dissolved in a freshly prepared citrate buffer (0.1 M, pH 4.5) was given to 12 normal Wistar rats to induce experimental DM1 (Furman, 2015). The control rats were injected with equal volumes of the vehicle. Blood samples were collected from the tail vein of all rats just before the streptozotocin injection and subsequently daily to monitor blood glucose levels using a blood glucose meter (Accu-Chek Performa; Roche Diagnostics, Indianapolis, IN, USA). Diabetic animals had mean random blood glucose levels of more than 24 mmol/l. Irisin was injected into the treatment groups at 5 µg twice a week for 6 weeks I/P prior to euthanasia allowing the animals to develop chronic complications of diabetes. Figure S1 shows the random blood glucose levels before the animals were euthanized. The animals were euthanized by CO2 overdose using commercially supplied compressed CO2 in cylinders fitted with Murex Saffire 300 Bar Argon/CO2 Mixed Gas Regulator Gauge by a vet and trained staff in the animal facility at CMHS. (100% CO2 was introduced to the chamber at a fill rate of 50% of the chamber volume per minute) followed by thoracotomy, 6 months after the induction of diabetes (Fig. 1).

Figure 1 A streptozotocin-induced rat model of type 1 diabetic osteopathy injected with a low dose of irisin.

Figure prepared using Canva software.

Blood and bones were collected for ELISA, PCR, western blotting, and imaging using micro-CT.

Only 15 out of 24 animals were used for this pilot study keeping in mind the 3Rs principle to see the effect of the low dose of irisin if any in treating diabetic osteopathy and the rest of the specimens were stored at −80 °C for future studies.

Power calculations were not carried out as it was a pilot study to test the effect of a very small dose of irisin on the trabecular bone that was not reported before in the literature.

The experimental animals were equally allocated to different groups at random for treatments and all procedures.

(a) Control+vehicle (Normal untreated NUT).

(b) Control+irisin (Normal treated NT).

(c) Diabetic+vehicle (Diabetic untreated DMUT).

(d) Diabetic+irisin (Diabetic treated DMT).

They were further subdivided for micro-CT, and bone turnover marker analysis at the end of the experimental period (n = 3 to 5) for each analysis. PI and research assistant were aware of the group allocation at different stages of the experiment.

Data acquisition using microcomputed tomography

The bone microarchitecture of the neck of the femur was examined non-invasively using a micro-CT (n = 3/Gp). The scanning and data analyses adhered to established guidelines for assessing bone microstructure in rodents, as outlined by Bouxsein et al. (2010).

The area of the Ward triangle (Bouxsein et al., 2010; Courtney et al., 1995) was scanned to detect any early changes in bone mineral density. Each specimen was scanned using a Nikon Metrology XT H225 (X-Tek Systems Ltd, Tring, Hertfordshire, UK) cone-beam μCT scanner operated at 65 kV, and 63 μA, with an exposure time of 1,000 ms. The geometric magnification produced a voxel dimension of ca. 23 μm for all the specimens. The software was set to optimize projections (typically 1,571), with two frames collected per projection. Noise reduction and beam hardening corrections were applied to the data.

To determine the trabecular bone microarchitecture in the femoral head and neck area, bone volume fraction (bone volume/total volume, BV/TV, %), trabecular bone thickness (Tb-Th, mm), trabecular bone separation (Tb-Sp, mm), and trabecular bone number (Tb-N, mm−1), the ratio of segmented bone surface to the total volume of the region of interest (BS/BV, mm−1), and bone mineral density (BMD, g cm−3) were measured using VG Studio Max 2.2 (Volume Graphics GmbH, Heidelberg, Germany) software. All trabecular bone microarchitectural measurements of the femoral head and neck area excluded the cortical bone as in the earlier study (Greenwood et al., 2018).

vTMD values were used to determine volumetric bone mineral density values (vBMD) according to: - vBMD = vTMD × BV/TV. vTMD refers to the density measurement restricted to within the volume of calcified bone tissue and excludes any surrounding soft tissue, whereas vBMD is the combined density in a well-defined volume (Estell et al., 2020).

A standard BMD phantom (QRM-microCT-HA, QRM GmbH, Moehrendorf, Germany) was used to quantify density within the micro-CT images. The phantom used consists of five cylindrical inserts of known densities of calcium hydroxyapatite (Ca-HA), Ca10(PO4)6(OH)2. Proprietary epoxy resin is uniformly filled as the base material. The BMD values of the cylindrical inserts were 1.13 gcm−3, 50, 200, 800, and 1,200 mgcm−3.

Real-time PCR analysis and western blots

Real-time PCR analysis and western blots were carried out in three to four randomly selected rats from each experimental group, to estimate the levels of SOST/sclerostin expression in bone samples at both transcriptional and translational levels. Real-time PCR was carried out by extracting RNA from tibiae by following the trizol method of RNA extraction (Kelly et al., 2014). The high-capacity cDNA reverse transcription kit (4368813; Applied Biosystems, Waltham, MA, USA) was used to synthesize cDNA from the extracted RNA. Real-time PCR analysis was performed using the TaqMan primers specific for SOST gene (4331182; Thermo Fisher Scientific, Waltham, MA, USA) detection and was normalized to β-actin (4331182; Thermo Fisher Scientific, Waltham, MA, USA) expression levels.

For western blots, a total protein was extracted from bone samples using a standard protocol. Briefly, bones were powdered in liquid nitrogen and extracted using 1X RIPA buffer containing protease and phosphatase inhibitors. Following centrifugation, the supernatant was collected and assayed by western blot analysis. A total of 20 µg proteins were separated in a 4–12% SDS-PAGE (M00654; Genscript, Jiangsu, China) and transferred to the PVDF (Polyvinylidene fluoride) blotting membrane. Following blocking with 5% milk in TBST (Tris Buffered Saline with Tween), the membrane was probed using a primary antibody against sclerostin (AF 1589, Mouse SOST/sclerostin antibody, 1:1,000 dilution in 5% milk in TBST) and Rabbit anti-goat IgG secondary antibody (Peroxidase conjugated, Cat# A4174, 1:6,000 in TBST; Sigma Aldrich, St. Louis, MO, USA). The blots were developed, and the images were captured on an X-ray film. The sclerostin western blot band intensities were normalized to the expression of GAPDH estimated by western blot analysis of the same samples using mouse monoclonal antibody against GAPDH (Sc-32233, Santa Cruz Biotechnology, 1:3,000 in 5% milk in TBST) and goat anti-mouse HRP-conjugated secondary antibody (ab205719, 1:5,000 in TBST) and shown as relative SOST expression.

Enzyme-linked immunosorbent assay

ELISA was carried out to estimate the bone turnover markers osteocalcin and C-terminal telopeptide (CTX1) levels in serum and bone samples in three to five randomly selected rats from each experimental group using a readymade kit from Abbkine Scientific (Osteocalcin, KTE1010153) and Cloud-Clone (CTX-1, CEA665Ra) respectively and following the standard manufacturer’s protocol. Briefly, 50 μl of the samples (for bone lysates, approximately 600 μg protein) or standards were applied to 96 well microtiter plates pre-coated with the ELISA capture antibody, mixed with 50 μl of 1:100 diluted biotin-conjugated competitor and further incubated for 1 h at 37 °C. The plates were washed thrice with the wash solution, incubated for 30 min with 100 μl of 1:100 diluted streptavidin-HRP, and washed five times with the wash solution. The plates were incubated with 90 μl of HRP substrate in the dark at 37 °C and the colorimetric reaction was quenched using a stop solution. The absorbance of the plate was measured at 450 nm spectrophotometrically (Tecan Infinite M200 Pro).

Statistical analysis

The data were analyzed using One-way or Two-way ANOVA with Turkey or Bonferroni post-test multiple comparison tests using commercially available software GraphPad Prism 9.0.0 for Windows (San Diego, CA, USA). Adjusted p-value (*p < 0:05, **p < 0:01). Data is presented as mean ± standard error (SE). The limited sample size used in this pilot study diminishes its statistical power, which is crucial for detecting real effects. Therefore the preliminary data presented here should be used to understand the trends and refine techniques for future experiments.

Results

Effect of irisin on blood glucose levels

The blood glucose level (mean ± SD) for nondiabetic/normal untreated (NUT) rats in Gp. I was 5.5 ± 0.7 (mmol/liter) and 4.4 ± 0.44 (mmol/liter) in nondiabetic/normal treated (NT) Gp. II. The blood glucose levels (mean ± SD) for diabetic rats were 25 ± 1.98 and 23 ± 3.8 in diabetic untreated (DMUT) Gp. III and diabetic treated (DMT) Gp. IV, respectively (Fig. S1). Treatment with irisin only marginally improved the blood glucose levels.

Trabecular bone morphometry using microcomputed tomography (micro-CT)

Data for all the measured trabecular bone structural parameters is presented in Table 1 as mean ± SE while Fig. 2 displays the 3D images of the micro-CT scans for each of the four experimental groups along with the plots depicting changes in various structural parameters of trabecular bone.

Table 1 Mean ± S.E between different groups related to trabecular bone parameters.

Mean ± S.E between different groups related to trabecular bone parameters obtained using micro-CT. Gp. I—normal un-treated/NUT, Gp. II—normally treated (NT), Gp. III—diabetic un-treated (DMUT), and Gp. IV—diabetic treated (DMT). Trabecular separation Tb-Sp Gp (I–III = p < 0.05: Trabecular thickness Tb-Th Gp (II and III; II–IV = p < 0.05): Trabecular number Tb-N Gp (III and IV = p < 0.05): bone volume/total volume BV/TV Gp (I–III; II and III = p < 0.05): bone surface density BS/BV Gp (II and III; II–IV p < 0.05): Bone mineral density BMD Gp (I–III; II and III = p < 0.05). n = 3/Gp.

Parameters	Mean ± S.E in the experimental groups	
NUT	NT	DMUT	DMT	
Tb-Sp (mm)	0.09867 ± 0.007781	0.1079 ± 0.01554
* with DMUT	0.1570 ± 0.008653
* with NUT	0.1137 ± 0.008182	
Tb-Th (mm)	0.1051 ± 0.007647	0.1189 ± 0.01297	0.0803 ± 0.008294
* with NT	0.0789 ± 0.002389
* with NT	
Tb-N (1/mm)	4.910 ± 0.08251	4.422 ± 0.1725	4.243 ± 0.2492	5.222 ± 0.2683
* with DMUT	
BV/TV%	0.5159 ± 0.03683	0.5255 ± 0.05855
* with DMUT	0.3376 ± 0.02096
* with NUT	0.4109 ± 0.01061	
BS/BV 1/mm	19.24 ± 1.478	17.19 ± 1.704	25.39 ± 2.427
* with NT	25.39 ± 0.7572
* with NT	
BMD g/cm 2	0.7527 ± 0.05921	0.7847 ± 0.1022
* with NUT	0.4580 ± 0.04238
* with NUT	0.5820 ± 0.02126	
Notes:

* p < 0.05.

Normal untreated NUT (Control+vehicle).

Normal treated NT (Control+irisin).

Diabetic untreated DMUT (Diabetic+vehicle).

Gp IV Diabetic treated DMT Diabetic+irisin.

Figure 2 Representation of 3D microarchitecture of the trabecular bone obtained by using the micro-CT.

Plots of changes in various structural parameters of trabecular bone in all experimental groups is shown. Representation of 3D microarchitecture of the trabecular bone at the proximal end of the femur is shown in frontal (A, C, E, and G) and cross-sectional (B, D, F, and H) images from four groups: (A and B) (Normal un-treated/NUT), (C and D) (Normal treated/NT), (E and F) (diabetic un-treated/DMUT), and (G and H) (treated/DMT) obtained by using the micro-CT. The image (I) is the magnified image of (A) to show the region of interest for frontal (red box) and cross-sectional (blue line) images. Plots of changes in various structural parameters of trabecular bone n = 3/Gp: (J) Trabecular separation (Tb-Sp) (K) Trabecular thickness (Tb-Th) (L) Trabecular number (Tb-N) (M) Bone volume/total volume BV/TV, (N) Bone surface density (BS/TV) (O) mean 1 (BMD), from NUT, NT, DMUT, DMT compared. p values are indicated in brackets.

The untreated diabetic group (DMUT) demonstrated an increase in the mean distance between trabeculae, resulting in larger marrow spaces (Table 1 and Fig. 2). The trabecular separation showed a significant 59% increase between the control group NUT (mean ± SE: 0.09867 ± 0.007) and DMUT (mean ± SE: 0.1570 ± 0.008). Treatment with irisin reduced the trabecular separation to 28% in the diabetic samples (mean ± SE: 0.1137 ± 0.008), although this change was not statistically significant (p > 0.05).

In terms of trabecular count, DMUT (mean ± SE: 4.243 ± 0.24) had a lower value compared to NUT (mean ± SE: 4.910 ± 0.082), although the difference was not statistically significant. Treatment with irisin resulted in a significant increase (p < 0.05) in the number of trabeculae in the diabetic samples (mean ± SE: 5.222 ± 0.268), with a recorded difference of 23%. Trabecular thickness decreased by 23% in DMUT (mean ± SE: 0.0803 ± 0.008) compared to the control NUT samples, and the irisin treatment did not show a significant improvement in this parameter (DMT: mean ± SE: 0.0789 ± 0.0023).

DM had a negative impact on bone volume fraction (BV/TV), as evident in the comparison between NUT, NT, and DMUT, with a significant decrease of 34.5% in DMUT compared to the untreated controls (NUT). Notably, the irisin treatment led to a 21.7% improvement in bone volume. Bone mineral density (BMD) exhibited a significant decrease in DM, with a statistically significant change of 39% calculated between the control NUT (mean ± SE: 0.7527 ± 0.05921) and DMUT (mean ± SE: 0.4580 ± 0.042) samples. The irisin treatment showed an increase of 27% in BMD (mean ± SE: 0.5820 ± 0.021) for the diabetic samples.

Effect of irisin on bone turnover markers

Bone formation decreased significantly in diabetes as indicated by the decreased osteocalcin levels in sera and bone samples in DMUT (Figs. 3A and 3B).

Figure 3 Plots of changes in bone markers in sera and bone tissue.

Plots of changes in bone markers in sera and bone tissue is shown (A–E) in all four groups (Normal un-treated NUT: Normal Treated NT: Diabetic untreated DMUT: Diabetic treated DMT) n = 3–5/Gp; F (n = 3–4/Gp): (A) Serum osteocalcin (ng/ml) (B) Bone osteocalcin (pg/ml) (C) Serum CTX1 (ng/ml) (D) Bone CTX1 (pg/ml). Relative SOST expression is shown by PCR (E), Western blot (F). p values are indicated in brackets. Error bars = mean ± SE.

Irisin treatment has anabolic action and it improved the osteoblastic activity reflected in raised osteocalcin levels, although the change was not statistically significant. Bone resorption as indicated by measuring CTX-1 in serum and bone samples indicates that resorption increases significantly in diabetes. Treatment with irisin further increased osteoclastic activity and this effect was significant in NT bone samples when compared with those of NUT (Figs. 3C and 3D).

We also observed that SOST levels were increased significantly in DMUT compared to NUT bone samples (Fig. 3F) p < 0.01 and were significantly down-regulated with irisin treatment in diabetic samples (p < 0.01) in both serum and bone samples (Figs. 3E and 3F and Datas S1A and S1B).

Discussion

DM1 is associated with poor bone health and a 6-fold increase in the overall incidence of hip fractures (Janghorbani et al., 2006; Janghorbani et al., 2007). Exercise improves many diabetic complications (Colberg et al., 2016). Physical activity stimulates the production of PGC-1α (peroxisome proliferator-activated receptor-γ co-activator 1α) in skeletal muscles, which in turn leads to the synthesis of FNDC5 (fibronectin type III domain-containing protein 5), a membrane protein abundantly found in skeletal muscles. Exercise increase the levels of irisin peptide, which is derived from the cleavage of its precursor protein FNDC5, as demonstrated in the research by Boström et al. (2012).

The research conducted by Faienza et al. (2018) revealed a significant inverse correlation between levels of irisin and the duration of diabetes. Another study found that the circulating irisin levels were lower in patients with diabetes when compared with healthy-matched controls (Tentolouris et al., 2018). The random blood glucose levels measured after the irisin treatment in controls and diabetic samples did not show any statistically significant improvement in our study. An earlier study by Duan et al. (2016) found that the therapeutic effect of irisin in lowering blood glucose is dose-dependent. The dosage (Duan et al., 2016) tested was much higher than what was used in this study. The non-significant decrease in blood glucose levels in our study can be attributed to the low dosage of irisin and the small sample size in this pilot study.

Colaianni et al. (2015) and Faienza et al. (2018) have shown that irisin is directly involved in bone metabolism, by promoting the differentiation of bone marrow stromal cells into mature osteoblasts. We specifically used a very small dose of irisin in this pilot study that has not been previously reported for bone tissue research. This decision was based on previous evidence showing that even a low dose of irisin, as low as 15 ng/ml, can increase AMPK levels in cells. Additionally, it has been demonstrated that 15 ng/ml is the observed serum level of irisin in diabetic rats after exercise (Formigari et al., 2022). Other studies have utilized doses of 50 and 100 ng of irisin to effectively stimulate significant increases in rodents (Kutlu et al., 2023). Furthermore, a recent study reported that irisin at a dose of 10 ng/ml can inhibit cell death and prevent mineral loss in bone tissue (Cariati et al., 2023).

Our study investigated the effect of DM1 on trabecular bone microstructure in the proximal femur obtained from mature male Wistar rats using a micro-CT. Furthermore, we examined the potential therapeutic effects of irisin in mitigating type 1 diabetic osteopathy induced by STZ. Additionally, the study evaluated changes in bone turnover markers after irisin treatment, including those specifically related to DM1.

Wistar rats are commonly used in animal research due to similarities in pathophysiologic responses between the human and rat skeleton, combined with the husbandry and financial advantages (Lelovas et al., 2008). Micro-CT is considered a gold standard technique for evaluating bone microstructure in small animal models. We successfully utilized micro-CT to analyze bone microarchitecture in type 2 diabetes in our earlier study Mohsin et al. (2019b).

Ward’s triangle is situated at the base of the femoral neck and is regarded as an area of minor resistance. The change in bone mineral density occurs early at Ward’s triangle; therefore, evaluation of bone mineral density in this area contributes to an understanding of femoral neck bone mass distribution and any imbalance is particularly important to assess the risk of bone fragility (Bouxsein et al., 2010; Furman, 2015).

DM adversely affects bone tissue making it porous and causing a decrease in bone volume/total volume, an increase in bone turnover (BS/BV), and a significant decrease in BMD (Chen et al., 2018).

Our study revealed several significant findings regarding the impact of diabetes on trabecular bone microstructure. We observed a notable increase in the distance between adjacent trabeculae, as indicated by increased trabecular separation (Tb-Sp), along with thinning of trabeculae in the DMUT group. Furthermore, we found an elevated bone surface-to-bone volume ratio (BS/BV) in the diabetic groups, suggesting increased osteoclast activity in diabetes. Although the number of trabeculae decreased in the DMUT group compared to the NUT group, the decrease was not statistically significant.

This study found that DM1 negatively affects bone volume fraction (BV/TV). Prior research (Boutroy et al., 2011; Ciarelli et al., 2000; Legrand et al., 2000; Milovanovic et al., 2012) noted lower BV/TV attributed to decreased trabecular number (Tb-N) and increased trabecular separation (Tb-Sp). Administering irisin increased the BV/TV by 21.7% and decreased Tb-Sp by (28%) in the DMT irisin-treated group as compared to DMUT. The number of trabeculae significantly increased with irisin treatment DMT. Reduction in BV/TV is a key structural alteration observed in osteoporotic bone, and it has been correlated with overall bone strength in various studies, including those conducted by Riggs & Parfitt (2005), Thomsen, Ebbesen & Mosekilde (1998), and Zhang et al. (2010).

A measure of bone mineral density (BMD, mg cm−3) is important in the evaluation of osteoporosis. Low bone mineral density along with poor bone quality is a risk factor for fragility fractures (Ciarelli et al., 2000; Marshall, Johnell & Wedel, 1996; Siris et al., 2001; Zhang et al., 2010). In this study, we observed that BMD significantly decreased in the untreated diabetic group of animals and irisin treatment improved the bone mineral density by 27%.

This study did not find a statistically significant change in the trabecular bone parameters in irisin-treated healthy animals in the control group. This is in agreement with a previously published study (Colaianni et al., 2015) which found no change in trabecular bone morphology related to Tb. Th, Tb-N, and Tb-Sp in mice treated with a low dose of r-irisin compared with the control mice. However, that study reported increased cortical bone mineral density and a positive effect on cortical bone geometry following irisin treatment (Colaianni et al., 2015). Nevertheless, a recent study of micro-CT analysis of femurs (Colaianni et al., 2017) showed that r-irisin maintained bone mineral density in both cortical and trabecular bone, and prevented a significant decrease of the trabecular bone volume fraction in hind-limb suspended mice. The thickening of the cortical bones after the irisin treatment is also evident in our experiments (Figs. 2D and 2H).

The alteration in the bone microstructure is attributed to changes in the remodeling cycle. Homeostasis in bone requires a balance between bone formation and resorption. Proper vascularisation is indispensable to maintain homeostasis. The impairment of blood supply to the bone tissue as occurs in diabetes could change the proliferation and differentiation of bone precursors in the bone marrow resulting in an altered bone remodeling cycle (Oikawa et al., 2010). RANK-ligand (RANKL) expressed by osteoblasts activates pre-osteoclasts to become mature osteoclasts through binding to receptor activator of nuclear factor-κB (RANK) receptors (Poole et al., 2005; Wijenayaka et al., 2011). Sclerostin, released by osteocytes has been reported to increase in DM1 (Hie et al., 2007; Kim et al., 2015). Our study showed that irisin treatment significantly decreased sclerostin levels in both normal and diabetic samples as shown in earlier studies (Colaianni et al., 2015; Klangjareonchai et al., 2014; Zhang et al., 2018). Sclerostin inhibits osteoblast differentiation and bone formation by antagonizing the canonical Wnt pathway (Delgado-Calle, Sato & Bellido, 2017). It also upregulates RANKL and downregulates OPG, leading to increased osteoclast activity and bone resorption (Poole et al., 2005). In osteoclasts, the expression of cathepsin K, TRAP (tartrate-resistant acid phosphatase), and carbonic anhydrase-2 proteins, involved in the remodeling of the extracellular matrix are upregulated by sclerostin (Wijenayaka et al., 2011).

The osteoblastic activity was estimated by measuring the osteocalcin levels in serum and bone samples. Our study also found that untreated diabetic samples had reduced osteocalcin levels, and treatment with irisin showed anabolic effects, by increasing osteocalcin release although the results were not statistically significant. Perhaps suppression of sclerostin in treated samples likely contributed to improved bone formation. The data obtained from this study is consistent with others which also demonstrated decreased bone formation in diabetes by the significantly decreased level of osteocalcin (Horcajada-Molteni et al., 2001; Li et al., 2005). Hyperglycemia in diabetes inhibits osteoblast proliferation, promotes osteoclast differentiation, decreases osteocalcin and OPG expression, and reduces bone mineral density. Irisin directly acts on osteoblasts, stimulating proliferation and differentiation through the p38 MAPK and ERK pathways (Qiao et al., 2016).

This study examined bone resorption by measuring carboxy-terminal collagen crosslinks (CTX-1) levels in both bone and serum samples. The findings were in line with similar research (Khan & Fraser, 2015; Qiao et al., 2016), showing a significant increase in bone resorption in DM1. (DM). Further, irisin treatment did not significantly affect the osteoclastic activity in the diabetic samples possibly due to the limited number of samples. A significant change was, however, recorded in the bones of normal rats as irisin treatment further increases the osteoclastic activity as shown by Ng et al. (2018). Irisin was shown in an earlier study to induce osteoclastogenesis by acting on integrin which, subsequently acts as the receptor for irisin on osteoclasts. Irisin-induced osteoclastogenesis led to the release of carboxy-terminal collagen crosslinks (CTX) and enhanced bone resorption (Kim et al., 2018).

To our knowledge, this is the first study to report the positive effect of irisin on the trabecular bone microstructure in DM1. Irisin treatment significantly improves the Tb. N and improves Tb. Sp, BV/TV, and BMD by 22–28%. The small change could be attributed to a very low dose of irisin and the small number of animals used in this pilot study. However, the study also found that low doses of irisin significantly decreased sclerostin, an anti-anabolic osteokine in diabetic osteopathy.

Limitations of the study

In accordance with the revised ARRIVE guidelines (Percie du Sert et al., 2020), this study was conducted as a pilot study due to the absence of evidence regarding the effect of the very small dose of irisin tested on trabecular bone in the existing literature. A decision was made to employ a small sample size which limits the statistical power of the study. Moreover, the effect of irisin on the trabecular bone is also controversial as Colaianni et al. (2015) were able to find irisin in a dose of 100 μg kg–1 has anabolic effects on cortical bone only and did not find any significant change in the trabecular bone. This project represents a collaborative effort involving diverse research groups dedicated to investigating the impact of irisin on various organs within the context of diabetes. The primary goal of the research presented here was to meticulously assess any potential therapeutic effects, if present, stemming from an exceptionally low dose of irisin, directed solely at the trabecular bone. Consequently, this study did not encompass an in-depth analysis of irisin’s influence on glucose homeostasis. We specifically tested a very small dose of irisin that has not been previously reported for bone tissue research. Due to the limited sample size and preliminary nature of the study, drawing definitive conclusions is not possible but the novel findings obtained provide a strong rationale for conducting future investigations with larger sample sizes to explore the effects of multiple doses and conduct more comprehensive experiments to analyze changes in bone microstructure and further elucidate bone biology.

Conclusions

The preliminary data obtained in this pilot study using a micro-CT analysis corroborates that DM1 deteriorates the trabecular bone microstructure in the proximal end of the femur which is only partially improved by irisin. Bone formation is adversely affected in STZ-induced type 1 diabetic osteopathy which is shown in this pilot study by decreased osteocalcin and increased CTX1 and sclerostin levels. Irisin is a regulator of bone remodeling by acting on all the key players of the bone remodeling cycle. Irisin significantly decreases sclerostin levels in diabetic rats which most likely promotes osteoblast differentiation and bone formation enhancing the trabecular bone quality. However, regarding trabecular bone parameters, statistically significant improvement with the irisin treatment is observed only in the trabecular number. Bone mineral density, bone volume fraction, and trabecular separation improved by 22–28% only and this could be due to the small sample size and a small dose of irisin used for this pilot study. Conversely, irisin also promotes osteoclastic activity and, therefore, would help to treat diabetic osteopathy where low bone turnover is the underlying pathology. However, the changes reported here with irisin treatment were marginal and further work with variable doses of irisin with a larger sample size is required to establish the role of irisin in type 1 diabetic osteopathy.

Supplemental Information

Supplemental Information 1 Arrive Guidelines Checklist.

Click here for additional data file.

Supplemental Information 2 Random Blood Glucose levels in all experimental animal groups.

Click here for additional data file.

Supplemental Information 3 Plots of trabecular bone parameters.

Mean ± S.E between different groups related to trabecular bone parameters obtained using micro-CT. Gp. I—normal un-treated/NUT, Gp. II— normally treated (NT), Gp. III—diabetic un-treated (DMUT), and Gp. IV—diabetic treated (DMT). Trabecular separation Tb-Sp Gp (I–III = p < 0.05: Trabecular thickness Tb-Th Gp (II and III; II–IV = p < 0.05): Trabecular number Tb-N Gp (III and IV = p < 0.05): bone volume/total volume BV/TV Gp (I–III; II and III = p < 0.05): bone surface density BS/BV Gp (II and III; II–IV p < 0.05): Bone mineral density BMD Gp (I–III; II and III = p < 0.05). n = 3/Gp.

Click here for additional data file.

Supplemental Information 4 Raw Data for ELISA, RT PCR, Western Blot and Micro CT.

Click here for additional data file.

We are grateful to the members of the animal house facility and Ms. Crystal D’souza and Ms. Sara Saeed Dewaib Rahmah Alhmoudi for animal handling at the College of Medicine and Health Sciences, United Arab Emirates University, UAE, Al Ain.

Additional Information and Declarations

Competing Interests

Author Contributions

Animal Ethics

Data Availability

The authors declare that they have no competing interests.

Sahar Mohsin conceived and designed the experiments, performed the experiments, analyzed the data, prepared figures and/or tables, authored or reviewed drafts of the article, administration, and approved the final draft.

Fiona Brock performed the experiments, analyzed the data, prepared figures and/or tables, authored or reviewed drafts of the article, and approved the final draft.

Suneesh Kaimala performed the experiments, analyzed the data, prepared figures and/or tables, authored or reviewed drafts of the article, and approved the final draft.

Charlene Greenwood performed the experiments, analyzed the data, prepared figures and/or tables, authored or reviewed drafts of the article, and approved the final draft.

Mohsin Sulaiman performed the experiments, authored or reviewed drafts of the article, and approved the final draft.

Keith Rogers performed the experiments, analyzed the data, authored or reviewed drafts of the article, and approved the final draft.

Ernest Adeghate conceived and designed the experiments, performed the experiments, analyzed the data, authored or reviewed drafts of the article, and approved the final draft.

The following information was supplied relating to ethical approvals (i.e., approving body and any reference numbers):

This study was approved by the Animal Research Ethics Committee of the College of Medicine and Health Sciences, UAE University.

The following information was supplied regarding data availability:

The raw data for all the experiments (trabecular bone parameters, Westersn, ELISA, PCR) and random Blood Glucose levels in all experimental groups are available in the Supplemental Files.

The Micro-CT data is now available at Cranfield’s repository: Brock, Fiona; Rogers, Keith; Mohsin, Sahar (2023). Micro-CT data investigating effect of irisin on trabecular bone in diabetic osteopathy. Cranfield Online Research Data (CORD). Dataset. DOI 10.17862/cranfield.rd.22734764.v1.

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
