# Peer review of "A pilot study: effect of irisin on trabecular bone in a streptozotocin-induced animal model of type 1 diabetic osteopathy utilizing a micro-CT"

_PeerJ, doi:10.7717/peerj.16278_

## Round 0.1 · original submission · Major Revisions

Dear Dr. Mohsin,

Your manuscript entitled “Effect of irisin on trabecular bone in a streptozotocin-induced animal model of diabetic osteopathy; a micro-CT study" which you submitted to PeerJ, has been reviewed by the editor and 3 external reviewers.

I regret to inform you that the reviewers have raised some significant concerns. In light of these concerns, further editorial consideration would not be possible without extensive and substantive revisions. In particular, the issues regarding the power of the study and the results presentation (including figures) must be adequately addressed. The reviewers’ comments are reported below.

If you decide to resubmit the revised version, please summarize all the improvements made in the new version and give answers to all critical points raised in the reviewers’ report in an accompanying letter. Copy and paste each and every reviewer's comment above your response.

Please note that resubmitting your manuscript does not guarantee eventual acceptance. Since the requested changes are major, the revised manuscript will undergo a second round of review by the same reviewers. I must emphasize that the acceptability of the revision will depend upon the resolution of the points raised by the reviewers.

Sincerely yours,
Stefano Menini

Reviewer 1 ·

Basic reporting

In this article, the authors investigated the effects of irisin administration in rats with type 1 diabetes. The article is clearly and well written , the reviewed and reported literature is sufficient, however, it is necessary to enrich the material section. and methods and improve the presentation of results to allow for acceptability.

Experimental design

The article describes a nice experiment in which the effect of irisin on the osteoporosis associated with type 1 diabetes. It is necessary to take care of some details in the methodology section:
1. It is necessary to indicate the ethical source for the euthanasia of mice with an overdose of CO2
2. It would be good to include a figure of the timeline to better undestood of the results
3. The authors say in line 158 "Only 24 animals were used for this pilot study keeping in mind the 3Rs principle to see the effect of irisin". Please define "3Rs principle"
4. Please include the oligo sequence used for SOST and beta actin for PCR
5. It is necessary to explain the procedures used for the extraction of bone proteins (line 209)
6. It is necessary to specify the amount of protein loaded for the ELISA (line 222)
7. There is no sample size calculation and the power of analysis has not been mentioned. The authors are required to justify the experimental n for the mcro ct analyses.

Validity of the findings

The article presents original evidence on irisin as a therapeutic target for osteoporosis associated with type 1 diabetes.
1. In the figures it is difficult to identify the items because the letters to indicate them (a, b, c, etc.) are very small
2. In all graphs please put the value of p in the brakets instead of the asterisks
3. Please add the densitometry for the results of the SOST WB
4. Would it be possible to have as a supplementary figure the glucose levels of the rats before and after treatment with Irisin?
5. Although in Table 2 the percentage of change is obtained and its statistics are made, it would be recommendable that Table 1 also indicate the significant differences.

Reviewer 2 ·

Basic reporting

This study aimed to investigate the effect of chronic administration of irisin on osteopathy induced by diabetes type 1 using an animal model. In the present study, the trabecular bone parameters were determined by micro-CT analysis, and some bone turnover markers were determined by ELISA, qPCR, and western blot. The authors concluded that chronic administration of irisin rescues the diabetes-induced loss of trabecular bone and decreases sclerostin levels. While this study is novel, it has major weaknesses that the authors should take into account.

Minor points and typos:

1. On line 112 the authors use de abbreviation “AGEs” without a previous description. It is strongly recommended to limit the use of abbreviations unless they are used frequently in the text.
2. It must be specified that the animal model is a type-1 diabetes model.
3. It is strongly suggested to change the label of the groups, it is difficult to follow them along the manuscript. Labeling the groups using their origin (e.g. Control + Veh) will be better.

Major points:
1. The manuscript is very poorly written, it seems to be a draft, but not a final version of the manuscript. The authors need to do better work to describe the results.
2. The figures are not showing the study's relevance. The authors must present figures in a manuscript format. They have a lot of space and all figures are mislabeled. It is highly recommended to make a panel combining 3D images and graphs showing the quantifications (instead of the tables). See other studies using microCT analyses. DOI: 10.1007/s10534-022-00421-5; DOI: 10.1038/s41598-020-72441-5.
3. It is suggested to eliminate Table 2 and use the data in the description of the results.

Experimental design

The experimental design is clear and well-argued.

Validity of the findings

The results are not clear and there is not an acceptable description of them.
The discussion is too long and on some points is ambiguous.

Reviewer 3 ·

Basic reporting

This paper by Mohsin et al is well written. There is extensive literature review and background reported in the introduction and the structure and quality of the figures/tables is very good.

Experimental design

The experimental design is based on a well defined and meaningful research question regarding the role of irisin in diabetic osteopathy. However there are some concerns regarding the design of this study:
1) This study might be underpowered (n=3-5) to show statistically significant changes that are now only trends (for example changes in trabecular separation and BMD that are affected by irisin but are not statistically significant). The authors should provide rationale for number of mice used and power calculations.
2) It is mentioned that the authors chose to treat with low doses of irisin, although the rationale for this is not provided. Also, why did the authors chose to treat for 6 weeks but animals were not sacrificed until 6 months post diabetes induction. The authors should elaborate on the irisin dose/duration chosen.
3) It is unclear to me why the authors did not report data on cortical bone properties but only chose to analyze trabecular bone, particularly as previous studies referenced by the authors have shown positive effects of irisin on cortical bone in other animal models.

Validity of the findings

1) Concerns about statistical power are a major concern for this study. Would recommend increasing number of mice analyzed as due to the low number, definitive conclusions cannot be made.
2) Another finding of the authors is that relative SOST expression by PCR showed a decrease between Groups I and II, whereas protein expression was increased between groups I and II. It would be good if the authors could speculate as to why findings from PCR and western blot are different.
3) Additionally, there is a lot of emphasis on the discussion regarding the effects of diabetes on trabecular bone which is already reported in literature and does not add any value to the paper. I would recommend focusing more the discussion and conclusions on the findings of this study as they relate to irisin, and shortening it considerably.

Additional comments

Please remove specific statistical calculations from Figure legends and add to the images (particularly for figure 2).

---

## Round 0.2 · Major Revisions

Dear Dr. Mohsin,

Thank you for your resubmission. I have now received the report from the reviewers.

The recommendations were mixed. One reviewer recommended acceptance, one recommended acceptance after "minor changes", and one recommended major revisions. Reviewer #3 is still concerned about the preliminary nature of the findings, the lack of statistical power, and the number of animals used.

I suggest considering discussing the preliminary nature of the results and the possible lack of statistical power as limitations of the study (option also suggested by Reviewer 1). According to Reviewers 1 and 3, the real effect of irisin on glycemic control is another topic that needs to be clarified and discussed. Finally, the number of animals actually used must also be clarified.

If you decide to resubmit a re-revised version of your manuscript, please copy and paste any comments from each of the reviewers above your response. If you feel any of their points are inappropriate, you are certainly free to provide rebuttal in your covering letter. You are also kindly requested to provide a complete tracked changes version of the manuscript in order to make it easier to verify that the required changes have been made, and a rebuttal letter with clear responses to each of the reviewer's comments.

Please note that resubmitting your manuscript does not guarantee eventual acceptance. I must emphasize that the acceptability of the revision will depend upon the resolution of all the points raised by the reviewers.

Sincerely,
Stefano Menini

Reviewer 1 ·

Basic reporting

The article is well written, the cited literature is sufficient. The data presented are interesting and provide insight into bone fragility caused by type 1 diabetes.

Experimental design

This work is a pilot study, so the rigor is adequate and sufficiently detailed

Validity of the findings

The results are novel for the understanding of bone fragility in type 1 diabetes.
It is noteworthy that nowhere in the manuscript is importance given to the decrease in glucose caused both in healthy and diabetic mice by the administration of Irisin. It has been reported that exercise improves glucose control and irisin production or its administration could represent a therapeutic target not only for bone fragility in diabetes, but also for metabolic control in diabetes. This could be added to the discussion and is important to mention the limitations of the study because it is a pilot study.
In the conclusions, it must be specified that it is a model of type 1 diabetes and make reference to the fact that it is a pilot study.

Additional comments

From the title it should be mentioned that it is a pilot study

Reviewer 2 ·

Basic reporting

The authors attend to all the suggestion points, the manuscript has been improved and the figures were fixed.

Experimental design

The experimental design is clear and well-argued.

Validity of the findings

All underlying data have been provided; to be a pilot study they are robust, statistically sound, and controlled.

Reviewer 3 ·

Basic reporting

The structure of the article remains similar to previous submission.
One of the recommendations was to shorten and make the discussion more focused and relevant to the results, however the authors have not achieved that as the discussion is still not focused on their findings.
I would suggest removing the text from lines 388-399 and making the discussion more focused.

Experimental design

Although the authors state in their rebuttal the reasoning for using a low dose irisin, they do not clarify as to the timing of irisin as it related to the timing of euthanasia. Do the authors believe that a small irisin dose for 6 weeks would be sufficient to reverse the effects of 18 weeks of uncontrolled diabetes on the skeleton? In other words, why did they chose to only treat for 6 weeks and why did they not treat from the onset of diabetes?

Minor points:
1. In methods, state that irisin was given for 6 weeks prior to euthanasia. The timeline, even though there is a figure now is not very clear.

2. What time points during the study where the blood glucose measurements that are reported in the manuscript? Were they prior to or during/after irisin was administered? It would be recommended to clarify and to also provide evidence that glycemia was not affected by irisin treatment.

Validity of the findings

The authors have made some improvements to the manuscript, however the main concern about preliminary nature of the findings and lack of power is not addressed under the statistical analysis section of the manuscript. Furthermore, in their manuscript the authors mention that they used 24 animals for this study, however there is data reported on only 12-15 animals for both microCT and ELISA studies under RAW data. Where is the data on the rest of the animals?
In the manuscript under materials and methods it is stated that 40 animals were obtained, which is misleading. Instead the authors should only state how many animals they used for this particular study AND are actually reported in the manuscript (which is 12-15 from what I can tell based on RAW data).
The validity of the findings is questioned because of the small sample and these results are very preliminary and therefore difficult to draw any conclusions from.

---

## Round 0.3 · accepted · Accept

Dear Dr. Mohsin,

Thank you for submitting a revised version of your manuscript. I am pleased to inform you that your manuscript is accepted for publication in PeerJ in its current form.

I thank all reviewers for their efforts in improving the manuscript and the authors for their cooperation throughout the review process.

Sincerely yours,
Stefano Menini

Reviewer 1 ·

Basic reporting

The authors heeded the suggestions

Experimental design

The authors heeded the suggestions

Validity of the findings

The authors heeded the suggestions

Additional comments

The authors heeded the suggestions